# Quantifying Biases in LLM-as-a-Judge Evaluations

**Magda Dubois** [* 1]  **Harry Coppock** [* 1]  **Mario Giulianelli** [1 2]  **Ole Jorgensen** [1 3]
**Timo Flesch** [1]  **Lennart Luettgau** [* 1]  **Cozmin Ududec** [* 1]

## Abstract

The evaluation of large language models (LLMs) is increasingly performed by other LLMs, a setup commonly known as "LLM-as-a-judge", or autograders. While autograders offer a scalable alternative to human evaluation, they are not free from biases (e.g., favouring longer outputs or generations from their own model family). Here we propose a statistical framework based on Bayesian generalised linear models (GLMs) that enables researchers to address their primary research questions (e.g., LLM capability or risk assessment), while simultaneously identifying, quantifying and mitigating various biases in their autograders. Our approach can be applied to various evaluation formats (e.g., absolute scores or pairwise preferences) and augments traditional metrics (e.g., inter-rater agreement) by providing precise uncertainty estimates and clarifying sources of disagreement between graders. This framework also enables efficient counterfactual simulations without costly re-evaluation (e.g., assessing agreement after removing systematic biases). We demonstrate these capabilities through simulated examples, with all methods available in an open-source software package. Overall, we introduce a novel framework for autograder evaluation which allows researchers to detect, quantify and correct for various biases in a systematic way.

## 1. Introduction

Imagine a typical scenario: a researcher, Florence, is assessing how well an LLM does on a given task. Outside of true/false questions, the outputs can be quite complex, e.g., open-ended answers, agentic trajectories or intrinsic preferences. Techniques have been developed to assess these outputs, e.g., scoring rubrics or collecting preferences. Due to the high stochasticity of LLMs, typical LLM evaluations require collecting a lot of samples meaning that manually scoring each response would be very time-consuming. Florence[2], like many researchers, decides to build autograders to automate this task. As she is interested in grading open-ended questions, she creates a rubric and prompts an autograder to apply it. Being a careful researcher, she wants to assess how well the autograders scores align with her own. As commonly done, she decides to assess this using an inter-rater agreement, e.g., Krippendorff's $\alpha$ (Tam et al., 2024; Bavaresco et al., 2025). She might get a value close lower than zero, indicating substantial disagreement between her and the autograders. But what does this mean? Is this just random noise or is there a way to explain this disagreement?

Recent studies suggest that such disagreement may not just be noise, as autograders can exhibit systematic biases. For instance self-bias, where LLM-based graders assign higher scores to responses generated by the same LLM family (Panickssery et al., 2024; Liu et al., 2024b), or more broadly to machine-generated content over human-written responses (Liu et al., 2023). Another common issue is length bias, where longer answers are preferred regardless of their actual quality (Zheng et al., 2023; Dubois et al., 2025). Additional biases include preferences for certain writing styles, answer structures, or the presence of certain keywords (Koo et al., 2024; Wang et al., 2024; Stureborg et al., 2024; Wu & Aji, 2025).

Through careful observation of the outputs, Florence might identify that the autograders consistently assigns lower scores than she does. She suspects that there is actually no fundamental disagreement on what constitutes good or bad responses, but rather slightly different scoring thresholds. One way to test this would be to adapt the scoring rubric to encourage higher scores and rerun the evaluation, but this approach would be resource intensive. A more efficient alternative is to simulate a counterfactual: what would the scores look like if we removed the systematic shift? By adjusting for this bias in the existing data and recomputing

---

[*]Equal contribution [1]UK AI Security Institute, London, UK [2]University College London, UK [3]University of Oxford, UK. Correspondence to: Magda Dubois <magda.dubois@dsit.gov.uk>.

*Proceedings of the $43^{rd}$ International Conference on Machine Learning*, Seoul, South Korea. PMLR 306, 2026. Copyright 2026 by the author(s).

---

[2]In tribute to the pioneering work of two Florence Nightingales in statistics: the 19th-century nurse who applied statistical methods to public health and the 20th-century statistician

Krippendorff $\alpha$ on the simulated scores, Florence can test her hypothesis without collecting new data. Doing this, she might find a higher value which would confirm that the apparent disagreement was largely due to a systematic shift rather than fundamental differences in quality assessment.

By performing such analyses, researchers can transform vague notions of autograder unreliability into precise and actionable insights about specific biases. To achieve this at scale, researchers need a framework that can decompose disagreement into interpretable components, quantify each bias with uncertainty, and predict how removing specific biases would affect evaluation outcomes without requiring costly re-evaluation. Our Bayesian GLM framework provides exactly these capabilities: (1) jointly modeling multiple bias sources (self-bias, length bias, grader severity, item effects) to identify which biases are present and their relative importance, (2) providing posterior distributions that quantify not just whether biases exist but their precise magnitude and uncertainty, (3) supporting both absolute scoring and pairwise preference formats to handle diverse evaluation setups, and (4) enabling counterfactual simulations that reveal how evaluation outcomes would change if specific biases were removed.

In the following sections, we will demonstrate this framework by addressing five common evaluation challenges. We also apply our framework to the publicly available MT-Bench dataset to illustrate how it can be used on real evaluation data and to highlight its practical advantages. The complete case study is reported in Section 4. All statistical models presented in this paper were implemented using HiBayES (Luettgau et al., 2025) and to facilitate wider adoption, are made available in the open-source package[3].

## 2. Related work

### 2.1. Bias detection methods

Recent work has developed various methods to detect and measure biases in LLM-as-judge systems. Some methods use ad-hoc interventions targeting specific types of biases (Zheng et al., 2023; Shi et al., 2025; Wang et al., 2024), while others focus on descriptive statistics (Bavaresco et al., 2025; Stureborg et al., 2024). More sophisticated approaches use a separate LLM to generate counterfactual inputs or responses for bias measurement (Ye et al., 2024; Chen et al., 2024). Our framework quantifies biases directly from existing evaluation data using statistical inference, without requiring additional model calls. It jointly estimates multiple bias types from a single dataset with full uncertainty quantification, and can be used either as a standalone alternative or as a complement to intervention-based methods.

### 2.2. Bias correction methods

Many existing debiasing methods rely on costly training of external models. These often pursue different objectives (e.g., Zhang et al. (2025a) focus on inter-judge disagreement) and depend on iterative self-correction modules (e.g., Yang et al. (2025)). Other approaches incur costs through prompt engineering (Zhao et al., 2024), iterative prompt refinement (Lyu et al., 2025), ensemble methods (Zheng et al., 2023), or re-ranking (Liu et al., 2025). The closest to our work are statistical approaches such as prediction-powered inference (Dorner et al., 2026), but these require ground-truth labels, which our method bypasses. Unlike these correction methods, our framework provides bias quantification without modifying the grading system and enables counterfactual analysis without re-evaluation, training, or ground-truth labels.

### 2.3. LLM-as-a-judge benchmarks

Benchmarks such as MT-Bench (Zheng et al., 2023), JudgeBench (Tan et al., 2025), and RewardBench (Lambert et al., 2024) have been developed to systematically evaluate LLM-as-judge performance. While these benchmarks document biases in specific models, our contribution is a generalisable statistical framework that can be applied to any evaluation dataset to quantify biases with uncertainty estimates. To further demonstrate the applicability of our method, we present a case study on MT-Bench in Section 4.

## 3. Methods

We begin by explaining how autograder scores can be compared to human scores and how this comparison can be integrated into an LLM evaluation analysis (Question 3.1). We then show how to assess whether the graders are biased towards certain models being evaluated (Question 3.2), how to quantify individual differences in the case of multiple graders (Question 3.3), and how to analyse item-level patterns (Question 3.4). Finally, we show how this framework can be applied to pairwise judgments settings, how to quantify intransitive (e.g., cyclic) preferences and how to assess whether graders have biases towards longer formats (Question 3.5). For a summary of the evaluation questions along with their corresponding formalisations, cf. Appendix A.1.

### 3.1. How do scores from an autograder compare to scores from an expert?

Florence needs to assess how well two LLMs do at answering open-ended questions. Because she is automating their grading with autograders, she essentially needs to answer two questions: 1) Is LLM A or LLM B better at answering open-ended questions? 2) Can my autograder reliably evaluate LLM responses relative to a human annotator?

---

[3]https://github.com/UKGovernmentBEIS/hibayes

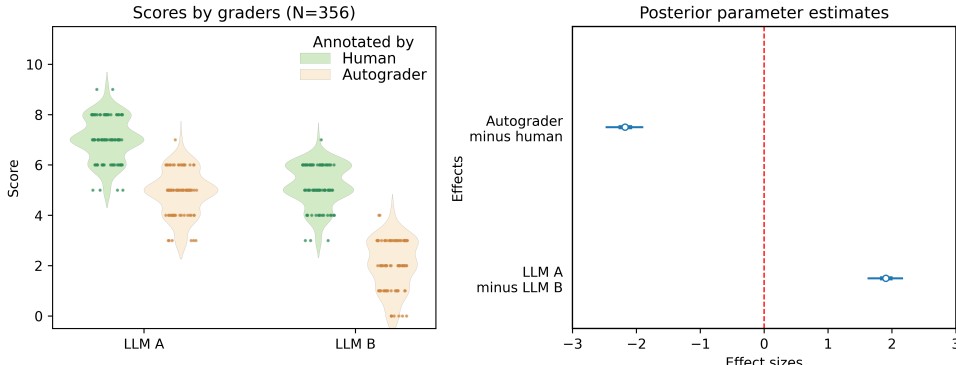

*Figure 1.* Addressing Question 1: how do scores from an autograder compare to scores from an expert?. Left panel: Simulated scores for LLM-generated answers graded by a human expert (Florence) and an autograder. Right panel: Posterior distributions of estimated effects. The horizontal blue lines represent 95% credible intervals. The dashed red vertical line indicates a null effect ($\beta = 0$). The coefficient for autograder minus human is negative, with a credible interval that does not include zero, indicating that the autograder tends to assign lower scores. The coefficient for LLM A minus LLM B is similarly positive, suggesting that LLM A receives higher scores than LLM B on average.

A GLM framework allows to answer these questions with a single analysis. Suppose that each LLM answered $N = 50$ items which were each graded by Florence and an autograder on a scale from 1-10 (simulated scores depicted on Figure 1).

GLMs extend linear regressions to handle non-normal outcomes while preserving the familiar regression structure. This is particularly useful as it allows researchers to include multiple predictors, control for potential confounders, and isolate the contribution of each variable to an outcome.

To answer the questions above, Florence can fit a regression model with an intercept $\beta_0$, which represents the overall average latent score, a coefficient $\beta_1$ which quantifies the effect of the grader, and coefficient $\beta_2$ which quantifies the effect of LLM A versus LLM B on the score (cf., linear predictor $\phi_i$ in Equation (1)).

$$\phi_i = \beta_0 + \beta_1 \cdot x_i^{\text{grader}} + \beta_2 \cdot x_i^{\text{LLM}}$$
$$\text{score}_i \sim \text{OrderedLogistic}(\phi_i, \boldsymbol{c}) \tag{1}$$

In this model, the linear predictor $\phi_i$ combines the effects of both the grader type and LLM identity. Because scores take discrete values from 1-10, we use an ordered logistic likelihood function. The linear predictor produces a continuous latent value, which the ordered logistic model maps to discrete 1-10 scores through estimated cutpoints $\boldsymbol{c}$. These cutpoints are estimated during model fitting along with the $\beta$ coefficients (cf. Appendix A.3). The variables $x_i^{\text{grader}}$ and $x_i^{\text{LLM}}$ encode the identity of the grader and the LLM respectively. Each variable takes a value of $+1$ or $-1$ (i.e., effect coding) to distinguish between the two levels (e.g., autograder vs. human, LLM A vs. LLM B).

After fitting, we can make two inferences based on the effect

sizes of the coefficients $\beta_1$ and $\beta_2$ (right panel of Figure 1):

1. The "Autograder minus human" effect is negative with credible intervals excluding zero, indicating that the autograder gives lower scores than the human expert

2. The "LLM A minus LLM B" effect is positive with credible intervals excluding zero, indicating that LLM A receives higher scores on average than LLM B.

Crucially, because the GLM accounts for both sources of variation simultaneously, Florence can confidently select LLM A for her task while remaining aware of the autograder's conservative scoring tendency. This illustrates how a GLM framework enables researchers to both answer substantive research questions and validate their evaluation methods within a single, principled analysis.

### 3.2. Do autograders favour their own generation?

Recent literature has raised concerns that autograders may demonstrate self-bias, a tendency to assign better scores to outputs generated by the same base model (Panickssery et al., 2024; Liu et al., 2024b; Koo et al., 2024) or outputs from models vs. humans (Liu et al., 2023). Similarly to above, a GLM framework allows to quantify autograder self-bias while evaluating LLMs.

Florence is concerned that her autograder (from model family A) might unfairly favour outputs from LLM A (also from model family A). To assess such self-bias, she uses a second autograder (from model family B). She wants to investigate whether responses from LLM A receive higher scores when graded by the autograder A compared to when graded by autograder B (and vice versa). The resulting data are shown in the left panel of Figure 2.

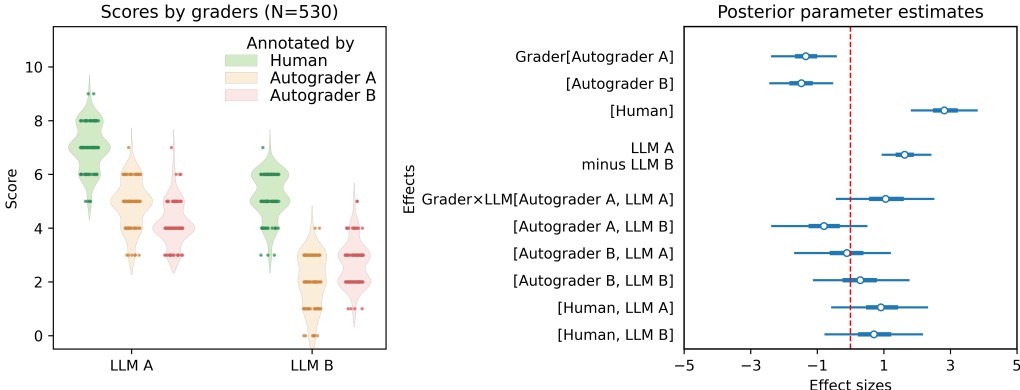

*Figure 2.* Addressing Question 2: Do autograders favour their own generation?. Left panel: Simulated scores for LLM-generated answers by LLM A and LLM B. Scores were given by a human expert (green) and autograders (yellow and red). Right panel: Posterior distributions of estimated effects from the GLM. The horizontal blue lines represent 95% credible intervals, and the dashed red vertical line indicates a null effect ($\beta = 0$). The grader effect $\beta_1$ shows how each grader deviates from the average score across all graders and LLMs. The LLM effect $\beta_2$ is positive, indicating that LLM A generally receives higher scores than LLM B. The grader-LLM terms $\beta_3$ represent a set of parameters (one for each grader-LLM combination). Autograder A seems to have a tendency to prefer LLM A vs LLM B, suggesting a potential self-bias.

We extend the previous model ( Equation (1)) by adding a term that captures whether specific graders systematically favour outputs from specific LLMs. This is implemented as a set of grader-LLM interaction effects[4], denoted by $\beta_3$.

$$\phi_i = \beta_0 + \beta_{1,g_i} + \beta_2 \cdot x_i^{\text{LLM}} + \beta_{3,g_i,\ell_i}$$
$$\text{score}_i \sim \text{OrderedLogistic}(\phi_i, \boldsymbol{c}) \quad (2)$$

As we now have more than two graders, the main effect coefficients $\beta_{1,1}, \beta_{1,2}, \beta_{1,3}$ represent each grader's deviation from the grand mean score, estimated using effect coding. The LLM variable still has two levels and is binary-coded as before (see Equation (1)). The interaction term $\beta_{3,j,k}$ represents a set of parameters estimated using index-based coding, with one distinct coefficient independently estimated for each grader $j$ and LLM $k$ combination.

After selecting the best-fitting model using model comparison techniques (Figure 8) in Appendix A.3), we examine the estimated effects (right panel of Figure 2. The interaction parameters $\beta_3$ show that Autograder A assigns somewhat higher scores to LLM A than to LLM B (positive vs. negative effects), suggesting potential self-bias toward outputs from its own model family (although this is not definitive as the credible intervals overlap with zero).

With these findings, Florence confidently answers her main question (LLM A performs better on open-ended questions), uncovers that the autograders assign lower scores than she

does, and identifies a potential self-bias from Autograder A.

### 3.3. Do autograders differ from human experts?

Florence might ask a few colleagues to help grade some of the responses (Human X, Y and Z in the left panel of Figure 3.), and try different an additional autograder (Autograder A, B and C in the left panel of Figure 3.).

She might then be interested in assessing: 1) whether autograder scores, on average, differ systematically from human scores, and 2) how much individual graders vary within each group.

To capture both group-level and individual-level differences, we can define what is known as a hierarchical GLM (cf. Equation (3)). In this model, each grader has their own scoring tendency ($\beta_{\text{grader}_i}$), drawn from a group-level distribution (human or autograder). This allows to estimate group-level means for humans versus autograders while also capturing how individual graders deviate from their group's average. Through partial pooling (sharing information across graders of the same type), the model makes efficient use of limited data, which is particularly helpful when some graders have few observations.

$$\phi_i = \beta_0 + \beta_{1,g_i} + \beta_2 \cdot x_i^{\text{LLM}}$$
$$\text{score}_i \sim \text{OrderedLogistic}(\phi_i, \boldsymbol{c})$$
$$\beta_{1,g_i} \sim \mathcal{N}(\mu_{\text{graderType}_i}, \sigma^2_{\text{graderType}_i}) \quad (3)$$
$$\mu_{\text{graderType}_i} \sim \mathcal{N}(0, 3)$$
$$\sigma^2_{\text{graderType}_i} \sim \text{HalfCauchy}(1)$$

---

[4]Strictly speaking, this is not a single interaction effect, but a set of parameters estimated independently using index-based coding (where each grader-LLM combination has a unique integer index). This allows direct comparisons across specific combinations rather than relying on a single coefficient.

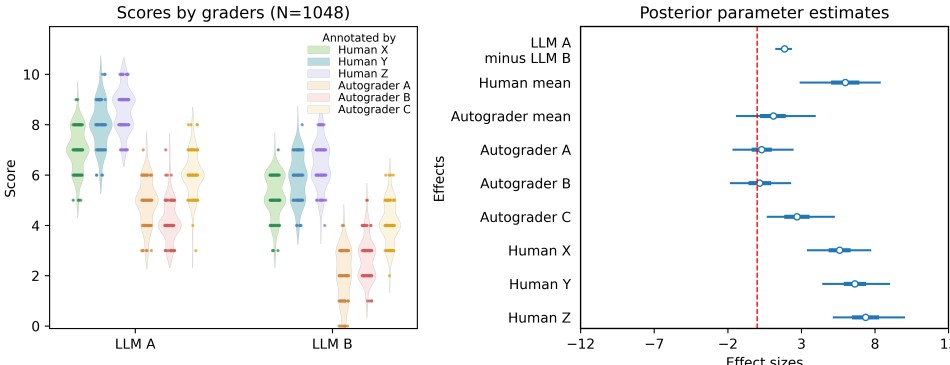

*Figure 3.* Addressing Question 3: Do autograders differ from human experts?. Left panel: Simulated scores on LLM A and LLM B answers, as graded by multiple human experts (green) and autograders (yellow and red). Right panel: Posterior distributions of estimated effects from the hierarchical model. The horizontal blue lines represent 95% credible intervals, and the dashed red vertical line indicates a null effect ($\beta = 0$). Individual grader effects show how each grader deviates from their respective group-level average (human or autograder). Group-level means for human and autograder graders ($\mu_{\text{graderType}}$) indicate that, on average, human graders assign higher scores than autograders.

As before, $\beta_2$ is a scalar coefficient applied to $x_i^{\text{LLM}} \in \{-1, +1\}$, which indicates whether the response was generated by LLM A or LLM B. Unlike before, $\beta_{1,g_i}$ represents the effect of the individual grader who assigned score $i$, and is drawn from a group-level distribution based on grader type (human or autograder). Specifically, $\beta_{1,g_i} \sim \mathcal{N}(\mu_{\text{graderType}_i}, \sigma^2_{\text{graderType}_i})$, where $\mu_{\text{graderType}_i}$ represents the average score tendency for each grader type, and $\sigma^2_{\text{graderType}_i}$ captures variability within each type. The prior distributions for the group-level means and variances ($\mu_{\text{graderType}}$ and $\sigma^2_{\text{graderType}}$) are specified in Appendix A.2. This hierarchical structure enables the model to estimate both the average difference between human and autograder scores and the variation among individual graders.

To formally assess whether this group-level difference exists, Florence should of course compare this hierarchical approach against a simpler flat model (cf. Figure 9 and Appendix A.3 for a model comparison). Here we select the hierarchical model to demonstrate how to examine both group-level differences between grader types and individual grader characteristics.

In the right panel of Figure 3 we see the individual grader effects ($\beta_1$; Autograder A-C and Human X-Z in the plot) and the group-level means ($\mu_{\text{graderType}}$; human mean and autograder mean in the plot). Using this method, Florence can confidently conclude that there is a general tendency for humans to give higher scores than autograders. Additionally, she can visualise individual-level differences and make informed decisions. For example, she might observe that Autograder C produces scores that are more closely aligned with those of the human graders. If consistency with human judgment is important, she may choose to use this autograder in future evaluations.

### 3.4. How do scores differ at an item level?

Florence now becomes interested in whether variation arises at the level of individual evaluation items (i.e., open-ended questions). She wonders whether some items consistently receive higher or lower scores, and whether graders agree more on certain items than others.

To answer these questions, she needs repeated responses for the same items. Until now, we have assumed that each data point corresponds to a different item. Lets instead imagine that Florences dataset consists of four items, with each model answering each item 25 times. The data split by items can be seen in the left panel of Figure 4 (different items are represented by violin plots of the same colour).

To answer Question 4, we extend Equation (1) by including two additional terms. The first term, $\beta_{3,m_i}$, accounts for a main effect of items, capturing whether some items receive systematically higher or lower scores. The second term, $\beta_{4,g_i,m_i}$, represents a grader-item interaction, allowing us to test whether particular graders behave differently on specific items.

$$\phi_i = \beta_0 + \beta_{1,g_i} + \beta_2 \cdot x_i^{\text{LLM}} + \beta_{3,m_i} + \beta_{4,g_i,m_i}$$
$$\text{score}_i \sim \text{OrderedLogistic}(\phi_i, \boldsymbol{c}) \tag{4}$$

The term $\beta_{1,g_i}$ captures the main effect of grader $g_i$, and $\beta_2$ models the effect of LLM identity (e.g., whether the response was produced by LLM A or B). As mentioned above, the new term $\beta_{3,m_i}$ represents the main effect of item $m_i$, which captures whether some questions tend to receive higher or lower scores overall. The final term, $\beta_{4,g_i,m_i}$, captures grader - item interactions and is implemented in the same way as the interaction term in Equation (2), i.e.,

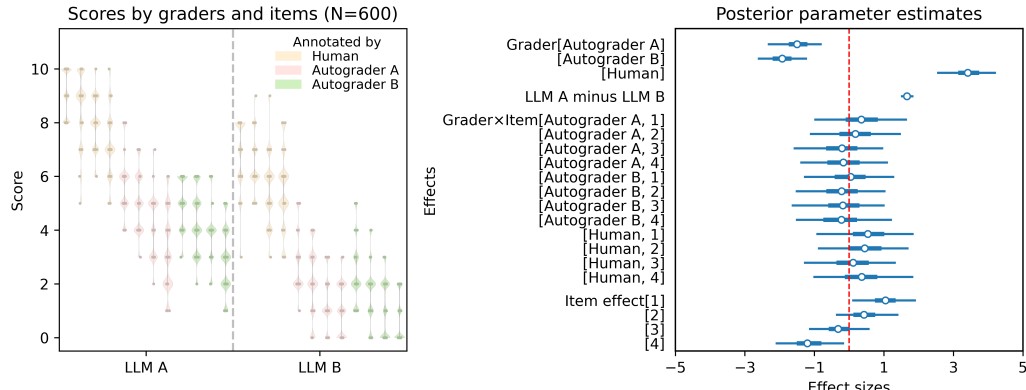

*Figure 4.* Addressing Question 4: How do scores differ at an item level?. Left panel: Simulated scores for each item (1-4), grouped by LLM and grader identity. Each cell shows the distribution of scores assigned by a given grader to responses from a particular model on a given item. Right panel: Posterior distributions of estimated effects from the item-level GLM ( Equation (4)). The plot shows main effects for grader and LLM identity (top), item main effects (bottom), and grader-item interactions (middle). Horizontal blue lines represent 95% credible intervals, and the dashed red vertical line indicates a null effect ($\beta = 0$). Item 1 has a strong positive effect, suggesting it consistently receives higher scores. In contrast, Item 4 has a negative effect, indicating that it receives lower scores. Grader - item interaction terms are small and uncertain, indicating no evidence of systematic grader disagreement on specific items.

a coefficient for combination. This allows to directly compare individual combinations and detect whether certain graders are more lenient or harsh on specific items. As before, all main categorical effects (grader, item) are encoded using effect coding, so that the resulting coefficients reflect deviations from the overall mean.

After fitting the model, Florence inspects the estimated effects (right panel of Figure 4).

1. Looking at the main effect of item ($\beta_3$ in Equation (4)), she observes that item 1 leads to higher scores and item 4 to lower scores. This suggests that the former is easier to answer, and the later more challenging.

2. Looking at the grader - item interaction term ($\beta_4$ in Equation (4)), she does not see evidence that specific graders differ on individual items.

From these results Florence concludes that while some items appear easier than others, grader disagreement is not concentrated on any particular question. The grader main effects reveals that humans consistently score higher than autograders, but Florence wonders about the consistency of their judgments beyond this systematic bias, i.e., do graders at least agree on which responses are relatively better or worse? Is the disagreement between graders due to fundamental differences in quality judgment, or merely due to correctable systematic biases?

This is where combining GLMs and inter-rater agreement metrics like Krippendorff's $\alpha$ becomes particularly valuable. While GLMs quantify systematic biases and $\alpha$ measures overall agreement, $\alpha$ alone cannot distinguish between random disagreement and systematic biases. For example, if

she were to compute $\alpha$ directly she would get $\alpha = -0.2$ (red cross in Figure 5), indicating substantial disagreement, but telling her little about why graders disagree or whether the disagreement is fixable.

The GLM framework offers two advantages here. First, by computing $\alpha$ on posterior samples from the fitted GLM, Florence can obtain not just a point estimate but a full distribution (blue in Figure 5). Second, and more importantly, GLMs enable counterfactual simulations. Florence can ask: "What would agreement look like if graders didn't have systematic biases?" To answer this, she removes each grader's estimated bias (the $\beta_1$ coefficients) from the linear predictor before mapping to categorical scores, then recomputes $\alpha$ on these bias-adjusted predictions.

This counterfactual $\alpha$ (green distribution in Figure 5) jumps to approximately 0.7, which is substantially higher than the observed $\alpha$. This reveals that most disagreement stems from systematic scoring differences rather than inconsistent judgments about quality. The wider green distribution reflects increased uncertainty after removing predictable grader variation.

Without the GLM framework, Florence would only know the agreement is poor. With it, she can decompose the disagreement into systematic biases (which can be corrected through calibration) versus fundamental disagreements about quality. The ability to simulate counterfactuals without collecting new data makes GLMs particularly valuable for understanding and improving evaluations.

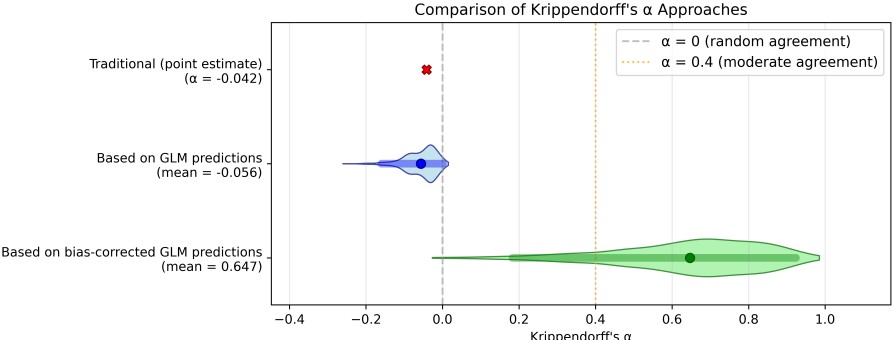

*Figure 5.* Posterior distributions of Krippendorffs $\alpha$ under different modeling assumptions. The red cross shows the traditional $\alpha$ computed directly from the observed scores, suggesting strong disagreement between graders. The blue distribution shows $\alpha$ values estimated from posterior simulations of a fitted GLM, incorporating uncertainty in the predictions. The green distribution shows $\alpha$ values after removing the main effect of grader identity, revealing what agreement might look like in a counterfactual scenario where graders do not differ systematically in scoring scale.

### 3.5. Do autograders favour longer outputs?

So far, we have focused on evaluation settings where graders assign absolute scores. However, many LLM evaluations rely on pairwise comparisons, where graders are asked to choose which of two outputs better satisfies a target criterion (e.g., correctness). The same statistical modeling framework can be applied in this setting, with the outcome modeled as a binary preference. We use such a setup to illustrate how pairwise comparisons can be modeled and to examine length bias, which has often been observed in pairwise evaluation settings. Of course, length bias can also be captured in absolute score setups similarly to other biases in previous sections.

Let's imagine that Florence wants to compare the quality of outputs generated by three different LLMs. She chooses a pairwise evaluation format, where each grader (e.g., herself or an autograder) is repeatedly shown two responses to the same prompt - each generated by a different LLM - and must select the better response. An example of such data can be seen in the left panel of Figure 6. Each bar represents a pairwise comparison (e.g., "LLM A vs. LLM B"), and its height reflects how frequently the first listed model (e.g., LLM A) was chosen. To model this binary outcome, we switch from the ordered logistic regression used previously to a binomial GLM with a logit link function. The outcome variable $y_i$ indicates whether the first model in the pair was chosen ($y_i = 1$) or not ($y_i = 0$), and we include a categorical effect in the model to denote the LLM pair being compared.

Florence's younger brother, always up-to-date with ML controversies, recently told her that some autograders may systematically prefer longer outputs even if those outputs are not of higher quality, a phenomenon commonly referred to as length bias (Zheng et al., 2023; Dubois et al., 2025). To capture this bias, she adds a continuous predictor capturing

the token-length difference between the two outputs. As there are two graders (herself and the autograder), which might have different biases, she computes one such predictor per grader. To test the existence of the length bias formally, she compares the model with and without this term (cf. Figure 10). For demonstration purposes, let's look at the model with grader-specific length bias here:

$$
\begin{aligned}
\text{logit}(p_i) &= \beta_0 + \beta_{1,\pi_i} + \beta_{2,g_i} + \beta_{3,g_i} \cdot x_i^{\text{lengthDiff}} \\
y_i &\sim \text{Binomial}(1, p_i) \\
\beta_{3,g_i} &\sim \mathcal{N}(\mu_{\text{lengthDiff}}, \sigma^2_{\text{lengthDiff}}) \\
\mu_{\text{lengthDiff}} &\sim \mathcal{N}(0, 0.5) \\
\sigma_{\text{lengthDiff}} &\sim \text{HalfNormal}(1.0)
\end{aligned}
\tag{5}
$$

where $y_i$ is a binary outcome indicating whether the first-listed LLM was preferred. The intercept $\beta_0$ is the overall tendency to prefer the first-listed model, $\beta_{1,\pi_i}$ captures pair-specific preferences (e.g., "LLM A vs. LLM B") where $\pi_i$ indexes the LLM pair, and $\beta_{2,g_i}$ captures grader $g_i$'s overall tendency to prefer the first-listed model. The predictor $x_i^{\text{lengthDiff}}$ is the token-length difference between the two responses. The grader-specific slope coefficient $\beta_{3,g_i}$ quantifies how sensitive grader $g_i$ is to length differences and is drawn from a hierarchical distribution with mean $\mu_{\text{lengthDiff}}$ and standard deviation $\sigma_{\text{lengthDiff}}$. Positive values of $\mu_{\text{lengthDiff}}$ indicate a preference for longer outputs. The hierarchical structure captures both the average length bias across graders and the variability among individual graders.

Importantly, once we have estimated the probability of choosing an LLM over another, we can compare these probabilities across pairs. This allows to identify rational (transitive) and irrational (intransitive) patterns of decision-making, such as cyclic dependencies (e.g., preferring A over B, B over C, but C over A). Such intransitivities exist in

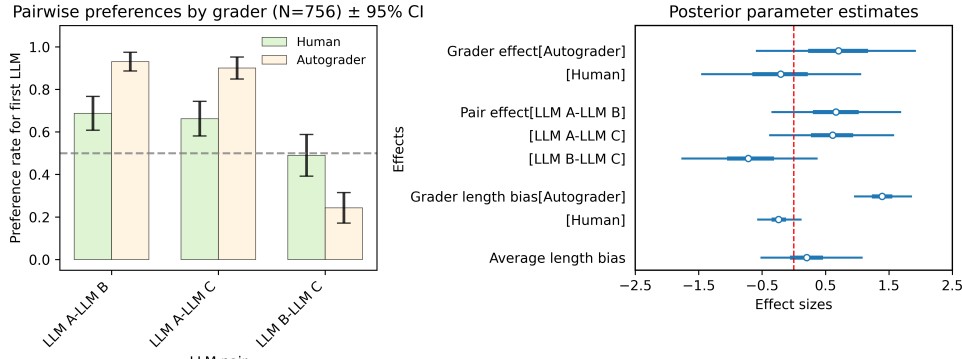

*Figure 6.* Addressing Question 5: Do autograders favour longer outputs?. Left panel: Proportion of pairwise preferences across three LLM pairs (A vs. B, A vs. C, B vs. C). Preference rate values represent the fraction of cases where the first listed model was selected over the second. Error bars represent 95% confidence intervals. Right panel: Posterior distributions of estimated effects from the GLM for pairwise comparisons (log-odds scale). Horizontal blue lines represent the 95% credible intervals, and the dashed red vertical line indicates a null effect ($\beta=0$). The pair effect terms represent the relative preference between specific pairs of LLMs, indicating which LLM is generally preferred. The grader length bias terms quantify each grader's sensitivity to token-length differences when making choices. A positive length bias indicates a preference towards longer outputs.

LLM evaluations (Xu et al., 2025). Traditional models like the Bradley-Terry model implicitly assume transitivity and thus cannot capture these cycles. Recent approaches have proposed either removing intransitivities from datasets (Yu et al., 2025) or explicitly quantifying them (Zhang et al., 2025b; Liu et al., 2024a; Zhao et al., 2024). GLMs naturally capture these intransitivities alongside grader biases and differences in LLM performance.

Inspecting the estimated effects in the right panel of Figure 6, we observe a positive effect for the grader-specific length bias parameter ($\beta_3$), particularly for the autograder. From this, Florence can infer that the autograder is more likely to select longer outputs, irrespective of their intrinsic quality, meaning that the autograder implicitly associated output length and perceived correctness. By explicitly quantifying such biases within the model, Florence can more reliably interpret differences in LLM rankings. For example, if LLM A wins most comparisons but consistently produces longer outputs, Florence might question: "Is LLM A genuinely better, or simply more verbose? Can I really trust the autograders judgements?". Additionally, by examining the estimated LLM pair parameters ($\beta_1$), she can verify whether the observed preferences follow a consistent ranking or if there are intransitive (cyclical) patterns. Here, the estimated parameters indicate a consistent ordering: LLM A tends to be preferred to LLM B and LLM C, and LLM B tends to be preferred to LLM C. This integrated statistical framework empowers her to disentangle and quantify these systematic biases and assess preference consistency, leading to deeper and more reliable conclusions.

## 4. Case study

To complement the simulated examples, we applied our full analysis pipeline to the publicly released MT-Bench dataset, which contains pairwise preferences from two graders. Following best practices, we constructed a sequence of models of increasing complexity:

- **Model M1**: Included only question effects (i.e., repeated measures). This model was added to illustrate that our method remains robust under misspecification.

- **Model M2**: Included grader effects (human or GPT-4).

- **Model M3**: Included question effects, grader effects, and two common biases present in MT-Bench: position bias and length bias. We also tested versions containing only one bias (omitted here for simplicity).

- **Model M4**: Extended M3 with a quadratic term for length. This model was added to demonstrate that the method naturally accommodates non linear structure.

- **Model M5**: Extended M3 with a three way interaction between grader, model, and position. This model was added to demonstrate how the framework handles more complex interaction effects.

- **Model M6**: Re-estimated M3 using tighter priors. This model was added to show that the results are not sensitive to prior specification.

- **Model M7**: Non hierarchical version of M3. Parameters are estimated independently rather than pooled. This allows to contrast hierarchical vs flat GLMs.

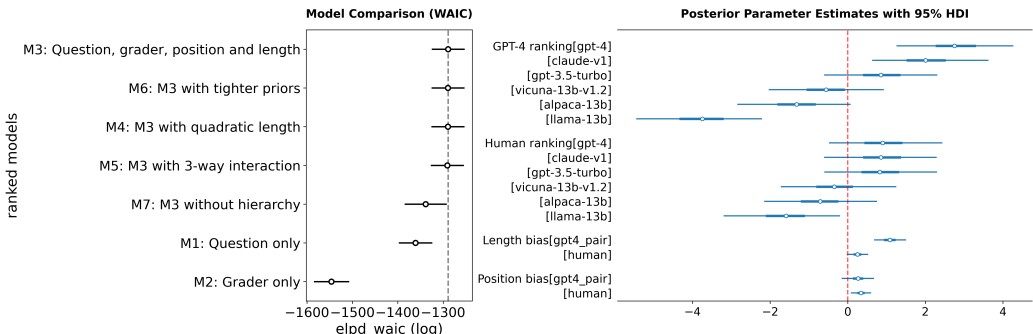

*Figure 7.* Model comparison (left) and forest plot of main effects (right) for the MT-Bench analysis.

A model comparison plot (LOO and WAIC) and a forest plot summarising the main effects are shown in Figure 7. As expected, M3 achieves the best balance between model fit and parsimony, and we therefore use M3 as the primary model when reporting results.

The forest plot highlights three effects of particular interest:

- **Length bias**: GPT-4 exhibits a substantially stronger length bias than the human grader (1.09 [0.68, 1.51] versus 0.25 [-0.02, 0.53]).

- **Systematic grader shifts**: GPT-4 shows positive shifts when evaluating outputs from GPT-4 itself (2.20 [0.29, 4.02]) and Claude (1.56 [-0.16, 3.46]), and a negative shift for Llama (-2.91 [-4.97, -0.86]). Human graders show more moderate patterns (0.34, 0.41, -0.75 respectively).

- **Position bias**: Human graders show mild position bias (0.34 [0.08, 0.61]).

From a single analysis, we can confidently say that one of our grader models exhibits a length bias, prefers outputs from certain models, and that human graders display a mild position bias. Importantly, we have quantified these effects precisely with full uncertainty estimates.

This demonstrates the practicality of our framework: multiple biases can be identified, quantified and jointly accounted for within a single principled analysis applied directly to real evaluation data.[5]

---

[5]The original MT-Bench study isolated individual biases (e.g., length, verbosity, position) through separate controlled experiments. A direct comparison is not possible as only a subset of the original data was publicly released; however, our analysis recovers the same qualitative findings: identical model rankings and detection of the same bias types.

## Conclusion

In this paper, we introduced a statistical framework for evaluating autograders using Bayesian GLMs. By jointly modelling the evaluation outcome and the scoring process, this approach enables researchers to assess both LLM performance and autograder behaviour within a single analysis. Importantly, it allows to jointly quantify multiple autograder biases directly from existing evaluation data with full uncertainty estimates. Through a series of examples, we followed the journey of a fictional researcher toward evaluating autograders. We used simulated data throughout to illustrate how to answer various questions in different evaluation settings.

Specifically, we showed how this framework can quantify various types of biases (e.g., self-bias and length bias), capture individual-level differences both among graders and items, and improve the estimation of group-level trends through hierarchical modelling. Crucially, the framework enhances traditional inter-rater agreement metrics in two ways: it provides uncertainty quantification through posterior distributions, and it enables counterfactual analysis to understand the sources of disagreement. Researchers can still compute familiar statistics such as Krippendorff's $\alpha$, Cohen's $\kappa$, or Kendall's $\tau$, but now with credible intervals and the ability to decompose disagreement into systematic biases versus fundamental differences in judgment. We also demonstrated the method's flexibility across different evaluation formats (e.g., absolute scores, pairwise comparisons).

The examples presented are by no means exhaustive. Many other applications and extensions are possible, and we hope this work provides a clear and practical starting point for researchers seeking to adapt the framework to their own evaluation scenarios. To support practical adoption, we have summarised common evaluation questions and their implementation in Appendix A.1, and provided a complete case study in Section 4. All statistical models presented in this paper can be found in the HiBayES open-source package[6].

---

[6]https://github.com/UKGovernmentBEIS/hibayes

## Impact Statement

This paper presents work whose goal is to advance the field of AI evaluations. There are many potential societal consequences of our work, none which we feel must be specifically highlighted here.

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

# A. Appendix

## A.1. Paper overview

*Table 1.* Overview of evaluation questions, their GLM-based implementation, and corresponding paper sections.

| QUESTION | GLM IMPLEMENTATION | IN |
|---|---|---|
| How does my autograder compare to humans? | Include both grader and LLM | Question 1 |
| Do my autograder(s) favour their own output? | Include an interaction between grader and LLM | Question 2 |
| Is there a general human vs autograder difference? | Use a hierarchical GLM with grader-level effects nested in grader type (e.g., human vs. autograder). | Question 3 |
| Are some graders more lenient or strict than others? | Estimate individual grader effects; inspect variation across graders. | Question 3 |
| Do some items receive higher or lower scores? | Include item as predictor; test whether some items are systematically easier/harder | Question 4 |
| Do graders disagree more on some items than others? | Include grader-item interaction; test for grader-specific scoring patterns | Question 4 |
| What is the uncertainty around inter-rater agreement metrics? | Simulate scores from the model and compute agreement (e.g., Krippendorff's $\alpha$) with uncertainty. | Question 4 |
| Do grader(s) favour longer responses? | Include token length (or token length difference) as a predictor. | Question 5 |
| Do my grader(s) exhibit intransitive | Estimate pairwise probabilities and | Question 5 |
| Is my grading scale well calibrated? | Inspect cutpoints from ordered regression to analyse spacing and interpretability of score intervals. | Appendix A.4. |

## A.2. Priors

Below are the priors used across the models described in this paper. They were selected to reflect weakly informative assumptions about effect sizes and score thresholds.

- Intercept: $\beta_0 \sim \mathcal{N}(0, 1)$

- Main effects of grader: $\beta^{\text{grader}} \sim \mathcal{N}(0, 1)$

- Main effects of LLM: $\beta^{\text{LLM}} \sim \mathcal{N}(0, 1)$

- Interaction effects: $\beta^{\text{interaction}} \sim \mathcal{N}(0, 1)$

- Group-level mean for grader type: $\mu_{\text{graderType}} \sim \mathcal{N}(0, 3)$

- Group-level standard deviation: $\sigma^2_{\text{graderType}} \sim \text{HalfCauchy}(1)$

- First cutpoint: $c_1 \sim \mathcal{N}(-4.0, 0.2)$

- Cutpoint differences: $c_j - c_{j-1} \sim \text{LogNormal}(-0.5, 0.3)$ for $j = 2, \ldots, K - 1$

To ensure ordered and well-separated cutpoints, the cutpoint differences are shifted by a small constant before summing: $\Delta_j = (c_j - c_{j-1}) + 0.3$.

## A.3. Supplementary figures: Model comparisons

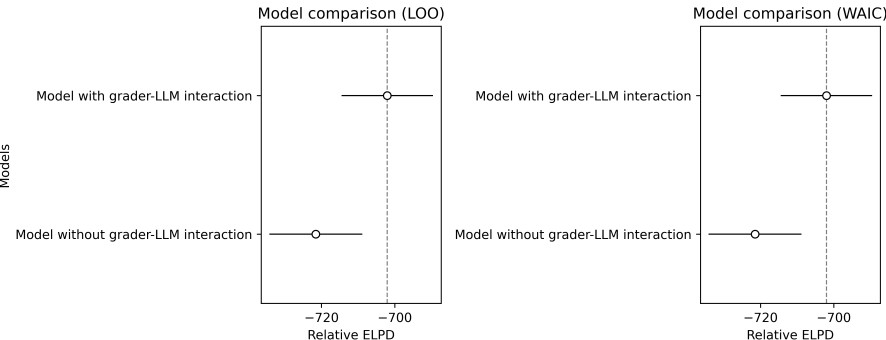

*Figure 8.* Model comparison for the statistical analysis in Question 3.2 (Do autograders favour their own generation?). Left panel: Leave-One-Out cross-validation (LOO) scores. Right panel: Widely Applicable Information Criterion (WAIC). Both metrics approximate the Expected Log Predictive Density (ELPD), a measure of predictive accuracy (higher values indicate better performance). Comparing models helps determine whether the added complexity of including an interaction term is justified by improved predictive performance. In this case, the model with the grader-LLM interaction ( Equation (2)) performs slightly better than the model without interaction, supporting a closer examination of potential self-bias effects.

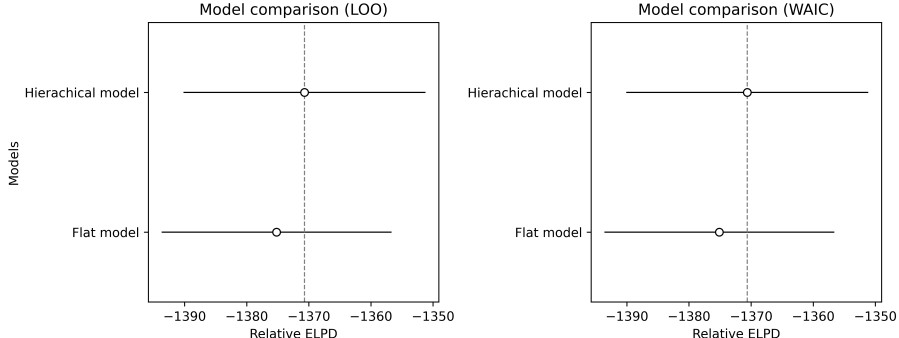

*Figure 9.* Model comparison for the statistical analysis in Question 3 (Do autograders differ systematically from human experts?). Left panel: Leave-One-Out cross-validation (LOO) scores. Right panel: Widely Applicable Information Criterion (WAIC). Both metrics approximate the Expected Log Predictive Density (ELPD), a measure of predictive accuracy (higher values indicate better performance). The models perform similarly, which is expected given that the data is simulated without an explicitly hierarchical structure. Here we choose the hierarchical model ( Equation (3)) to demonstrate how to interpret its parameters. In practice, when models perform similarly, researchers should favour the simpler model unless theoretical or interpretability considerations justify the added complexity.

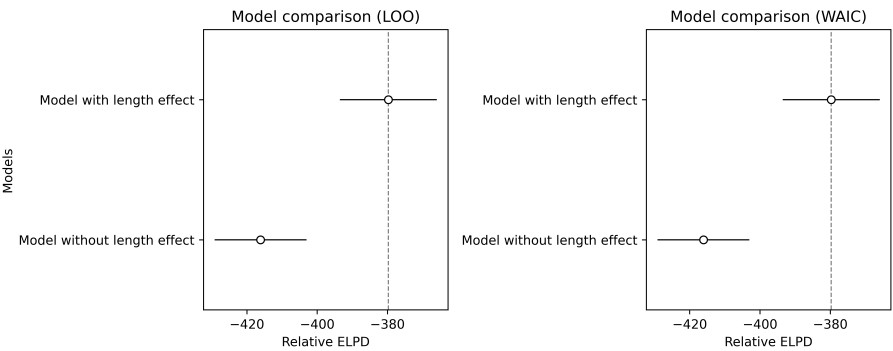

*Figure 10.* Model comparison for the statistical analysis in Question 5 (Do autograders favour longer outputs?). Left panel: Leave-One-Out cross-validation (LOO) scores. Right panel: Widely Applicable Information Criterion (WAIC). Both metrics approximate the Expected Log Predictive Density (ELPD), a measure of predictive accuracy (higher values indicate better performance). Comparing these models tests whether including a length-effect term ( Equation (5)) significantly improves predictive performance. The model including the length effect clearly outperforms the simpler model without it, justifying a closer investigation into grader-specific length biases.

*Table 2.* Example of a rubric score that a researcher might create to grade open-ended question.

| POINTS | DESCRIPTION |
|---|---|
| 1 | Completely off-topic or no relevant content. |
| 2 | Minimal response with no clear concepts or severe confusion. |
| 3 | Mentions a relevant idea but largely undeveloped or inaccurate. |
| 4 | States one relevant concept with limited clarity or major misconceptions. |
| 5 | Mentions key concepts but lacks depth and contains notable flaws. |
| 6 | Covers some key concepts with partial accuracy and development. |
| 7 | Addresses key concepts clearly, with minor omissions or inaccuracies. |
| 8 | Explains most important concepts with accuracy and reasonable depth. |
| 9 | Thorough and accurate response with clear development and insight. |
| 10 | Complete responses with deep understanding and insightful connections. |

### A.4. Supplementary question: Is the grading scale well calibrated?

In this section we explore the grading process, and why we chose an ordered logistic regression model. To ensure consistent evaluation, Florence needs to establish standardised scoring across all graders. One simple approach would be to instruct everyone (humans and autograders) to count the number of relevant keywords from a predefined list. Besides obvious issues (e.g., synonyms need to be accounted for), this approach is not scalable as it require building explicit keyword lists for each open-ended question. Florence, as often done in practice, will instead develop a grading rubric that can be applied to each question. She might come up with something like in Table Table 2.

Having an ordinal scale, instead of just a description, is of course very useful. But this scale is not as simple to interpret as an interval scale (e.g., temperature). If Florence wants to make claims like "autograder A increases the score by 1 point," she needs to be aware of whether the intervals between categories are equivalent. If they are not (which is often the case with ordinal scales), a 1-point increase will mean something different depending on where it occurs on the scale. For example, moving from 5 to 6 might represent crossing some fundamental threshold, while moving from 9 to 10 might just be a small qualitative improvement between two already good responses.

In an ordered logistic regression model, the scores are considered individual categories with a meaningful order. The model maps the observed scores onto a continuous latent scale and, if necessary, can estimate the category boundary values (called cutpoints) on this latent scale. By examining the spacing between these cutpoints, we can determine whether the intervals in our grading scale are equivalent across the range. If they are not, we can make inferences from the distances between cutpoints. For example, if some cutpoints are widely spread, it could indicate that we are capturing an important threshold (i.e., substantial changes in the underlying latent score is required to move between categories), that there is a gap in the

*Table 3.* Cutpoints and interpretations for score intervals

| CUTPOINT | VALUE | RANGE | SCORE | THRESHOLD |
|----------|-------|-------|-------|-----------|
| c1 | -4.07 | – | 0–1 | Starting point |
| c2 | -3.25 | 0.82 | 1–2 | Narrow |
| c3 | -2.39 | 0.86 | 2–3 | Narrow |
| c4 | -1.28 | 1.11 | 3–4 | Moderate |
| c5 | -0.20 | 1.08 | 4–5 | Moderate |
| c6 | 1.29 | 1.49 | 5–6 | Wide |
| c7 | 3.11 | 1.82 | 6–7 | Wide |
| c8 | 4.71 | 1.60 | 7–8 | Wide |
| c9 | 5.60 | 0.89 | 8–9 | Narrow |
| c10 | 6.22 | 0.62 | 9–10 | Narrow |

measurement scale or that graders are reluctant to use certain portions of the scale. Conversely, if some cutpoints are close together, it could indicate that the scale is very sensitive in that region (i.e., small changes in the latent score result in different observed scores), that models have similar latent abilities in that "region" or that the scale contains redundant categories.

Building great scales is by no means an easy task. It is a well-established challenge that has been thoroughly studied in the field of psychometrics, but is beyond the scope of the presented paper. However, using a GLM with an ordered logistic regression is a useful tool to examine the properties of a given scale and can help identify areas that require caution during interpretation.

Going back to Florence, to better capture differences in model capabilities, she decides to examine her grading scale. To do this, she can look at the learned cutpoint parameters of the ordered logistic regression. Taking Equation (1) for example, we can write the cumulative probability more explicitly as:

$$
\phi_i = \beta_0 + \beta_1 \cdot X_i^{\text{grader}} + \beta_2 \cdot X_i^{\text{LLM}}
$$
$$
p_{ij} = \text{logit}^{-1}(c_j - \phi_i)
$$
(6)

where $\phi_i$ is the linear predictor, representing the location of response $i$ on an unobserved latent scale. This latent scale is assumed to underlie the observed ordinal scores (e.g., 1-10). The cutpoints $c_j$ divide the latent scale into discrete intervals corresponding to the observed score categories. The probability $p_{ij}$ represents the cumulative probability that the score assigned to response $i$ is less than or equal to category $j$, and is calculated as the inverse logit of the difference between the cutpoint $c_j$ and the linear predictor $\phi_i$.

Conceptually, this means that the probability of scoring at (or below) a certain grade j (on the observed scale) is calculated by comparing the linear predictor to the cutpoint (on the latent scale). These cutpoints are the mechanism through which ordered logistic models maintain the ordinal structure of the data. They were therefore naturally present in the previously discussed models, but were not mentioned as the focus was on the predictor variables.

Once Florence fits this model, she can analyse the inferred cutpoint values. Those can be found in Table Table 3. She observes that the distance c1-c2, and c2-c3, is below 1 unit, whereas the distance c6-c7 and c7-c8 is above 1.5 units. This large jump suggests that her scale lacks sensitivity around c6-c8 (i.e., many performance scores are clustering there). From this, she could decide that she wants to better capture difference in that region and therefore adds some intermediate categories in this area.

