# OpenReview forum: "Quantifying Biases in LLM-as-a-Judge Evaluations"
_ICML.cc/2026/Conference — ICML 2026 regular_

### Official Review · Reviewer_Pmyq · 2026-02-25

**Soundness:** 3
**Presentation:** 3
**Significance:** 3
**Originality:** 3
**Overall Recommendation:** 5
**Confidence:** 5

**Summary:**

This study outlines a core issue in modern LLM evaluation: the widespread use of LLM-as-a-judge introduces systematic, poorly understood biases that are not captured by standard agreement metrics. The paper studies a broad theme by proposing a unified Bayesian generalized linear model (GLM) framework to jointly evaluate model performance and diagnose autograder biases (e.g., grader severity, self-bias, length bias, item effects) across absolute and pairwise evaluation settings. The framework provides uncertainty quantification, supports counterfactual debiasing analyses without re-evaluation, and is demonstrated through extensive simulated examples and a real-world case study on MT-Bench, with an accompanying open-source implementation.

**Compliance With Llm Reviewing Policy:**

Affirmed.

**Key Questions For Authors:**

Sensitivity to misspecification:
How robust are the bias estimates and counterfactual conclusions when the true data-generating process deviates from the assumed GLM structure (e.g., non-linear or prompt-specific effects)?

Scalability:
How does inference scale with many graders, models, and interaction terms in large benchmarks, and what are the practical computational limits?

Decision-making guidance:
How should practitioners act when bias credible intervals overlap zero (as in some self-bias examples)? Are there recommended thresholds or decision rules?

Comparison to simpler baselines:
In practice, when does this framework provide clear advantages over simpler post-hoc normalization or regression-based debiasing?

**Limitations:**

yes

**Strengths And Weaknesses:**

Strengths

Soundness:
The methodology is technically sound and grounded in well-established Bayesian GLMs and hierarchical modeling. Model specifications, priors, and likelihood choices are appropriate for ordinal and pairwise data. Claims are supported by clear simulations and a realistic MT-Bench case study. The use of posterior predictive checks, model comparison (LOO/WAIC), and counterfactual simulations is rigorous and well motivated.

Presentation:
The paper is clearly written, well structured, and pedagogical. The running example effectively guides the reader through increasingly complex models. Figures are informative, and the appendix provides sufficient detail for reproducibility. The positioning with respect to prior work on LLM-as-a-judge bias is thorough and fair.

Significance:
The work addresses a highly relevant and timely problem for the ML community. As LLM-based evaluation becomes standard, a principled way to quantify and correct evaluator bias is practically important. The framework is general and could influence how future benchmarks and evaluations are designed and interpreted.

Originality:
While the individual statistical tools are not novel, their integration into a unified, flexible framework for autograder evaluation is original and valuable. The counterfactual analysis of inter-rater agreement and the joint treatment of multiple bias sources meaningfully advance current practice beyond ad-hoc bias probes.

Weaknesses

Empirical validation:
Much of the main paper relies on simulated data; the real-data validation is confined largely to MT-Bench. Additional diverse real-world case studies would strengthen confidence in practical robustness and adoption.

Scope of bias coverage:
The framework models biases that are linear/additive in the predictors. More complex forms of bias (e.g., prompt-dependent or semantic biases) are acknowledged but not addressed, which limits completeness.

Accessibility:
Despite good exposition, the paper assumes substantial familiarity with Bayesian modeling. Some readers may find the statistical depth a barrier to adoption without higher-level guidance or defaults.

---

### Official Review · Reviewer_rS1d · 2026-03-09

**Soundness:** 1
**Presentation:** 1
**Significance:** 1
**Originality:** 1
**Overall Recommendation:** 2
**Confidence:** 4

**Summary:**

This paper considers the problem of quantifying biases in LLM-as-a-judge systems. The authors propose to use Bayesian generalized linear models (GLMs) to answer questions like "How do scores from an autograder compare to scores from an expert?" and "Do autograders favour their own generation?". The authors show a series of simulated examples to demonstrate how to use their framework.

**Compliance With Llm Reviewing Policy:**

Affirmed.

**Final Justification:**

The authors did not answer other reviewers' questions, nor did the other reviewers post any new comments. Therefore, I maintain the opinions in my Rebuttal Acknowledgement.

**Key Questions For Authors:**

None

**Limitations:**

See **Strengths And Weaknesses**

**Strengths And Weaknesses:**

Strengths:
1. The proposed framework provides a unified modeling perspective on quantifying biases. The core idea is that a single regression model can answer all sorts of bias-related questions.
2. The framework is reasonable, with root in classical statistical measurement.

Weaknesses:
1. There is nothing new methodologically. The Bayesian GLM models coupled with posterior inference is textbook statistics.
2. The paper presents simulated examples to demonstrate how to use their framework, without any real data or actually calling the LLMs. Therefore, the paper does not offer new empirical insights.
3. The paper offers no insight on how the priors or parameters should be chosen in practice.
4. It is not clear why Bayesian GLMs have clear advantages over simpler analysis such as paired t-tests. The more complex models also need more justification like well-specified priors and parameter identifiability.
5. The writing is poor. Notations are often not defined.
6. The authors claim that existing statistical approaches like prediction-powered inference requires ground-truth labels, which their method bypasses. However, it appears that the GLMs require human graders' scores.

---

> ### Author Rebuttal · Authors · 2026-03-30
>
> We thank the reviewer for their feedback. We would like to clarify two important misunderstandings in the reviewers comments: we do have real data in the paper (weakness point 2), and our framework does not require human graders (weakness point 6). Also, there are clear advantages of using GLMs over paired t-tests (weakness point 3). We explain these in detail below:
>
> ## About weakness point 2:
> *"The paper presents simulated examples to demonstrate how to use their framework, without any real data [..]"*
>
> The paper includes a full real data analysis in Appendix A.5 (the publicly released MT-Bench dataset). In there we fit a few statistical models to pairwise preferences from human and GPT-4 graders and show how from this one analysis we discover a length bias in GPT4, a position bias in human graders, and grader-specific shifts. Importantly, our pipeline incorporates model comparison, which shows that the added complexity of specific bias terms is supported by the data and improves predictive fit. In fact, this is one of the practical advantages of the framework over simpler ad hoc analyses: it allows researchers to ask not only what effects are present, but also whether the corresponding model components are justified. The analysis also provides full uncertainty quantification over these effects. We suspect that the misunderstanding about the absence of real data was due to the fact that the MT-Bench analysis is presented in Appendix A.5 rather than in the main paper. We would be happy to move it into the main text in the final version.
>
> ## About weakness point 6:
> *"[..] the GLMs require human graders' scores"*
>
> We would like to clarify that having human generated scores is not a requirement of the framework, it was just used as an example. Human scores are one possible grader source in some analyses (e.g., when comparing autograders to humans), but the framework can of course also be applied to datasets containing only autograders or only pairwise preference data (depending on the setup). More generally, our point in Section 2.2 is that, unlike prediction-powered inference approaches, our method does not require external ground-truth labels to quantify grader biases from the evaluation data itself.
>
> ## About weakness points 4 and 3:
> *"It is not clear why Bayesian GLMs have clear advantages over simpler analysis such as paired t-tests. [..]" and "[..] no insight on how the priors or parameters should be chosen [..]"*
>
> Regarding the question of why Bayesian GLMs are preferable to simpler analyses such as paired t-tests: our framework works for a wide range of outcome (e.g., ordinal, binary, pairwise), supports multiple simultaneous predictors and interactions, provides partial pooling in hierarchical settings, and enables counterfactual simulations (e.g., recomputing agreement after removing estimated systematic biases). These features are central to the value of the framework and are not handled naturally by a paired t-test.
>
> The question of which additional parameters to include in practice is exactly where model comparison is useful: researchers can compare different model specifications (i.e., different parameters) and retain added terms only when they improve predictive fit. By doing this, the choice of model components is empirical rather than fixed a priori.
>
> In Appendix A.2 we state that we use weakly informative priors, following standard practice in applied Bayesian modeling. We would be happy to expand this section and provide more concrete guidance on typical prior values for different distributions. As a note, while prior choice is important, with sufficient data the main conclusions are often driven primarily by the likelihood. We demonstrate this in the MT-Bench case study (Appendix A.5) where we fit models with different priors (models M3 vs M6) and find that the predictive fit (and therefore the conclusion) is not affected.
>
> ## About weakness point 1:
> *"There is nothing new methodologically. The Bayesian GLM models [..] is textbook statistics"*
>
> We appreciate the reviewer's concerns about novelty. The contribution of our work is the formalisation and introduction of these tools to LLM-as-a-judge evaluation and their integration into a unified framework. Our framework covers multiple common sources of bias, allows for principled uncertainty quantification, and model comparison based on predictive fit to identify the data generating process, and counterfactual analysis within a single workflow. This workflow and approach is new to the field of AI evaluation and will help increase the reliability and robustness of the statistical inferences drawn and the scientific findings reported. We make this concrete by specifying which model structures are appropriate for common evaluation setups (e.g., ordinal scoring, binary pass/fail, pairwise preferences) and which bias terms are relevant to consider in each setting. Of course the framework is also flexible to other sorts of setups and biases.

---

> > ### Author Rebuttal · Reviewer_rS1d · 2026-04-03
> >
> > I thank the authors for the detailed response. I appreciate the pointer to the MT-Bench case study in the appendix. I had overlooked that section in my initial read, and I apologize for that. I agree that the paper is stronger with this addition, and I do think the framework has potential and is useful. However, my concern is that it still does not meet the acceptance threshold I would expect for this venue, and I remain leaning toward rejection, so I will maintain my score. My reasons are as follows:
> >
> > 1. **I find the overall contribution limited.**
> >
> > There are at least two ways I think a paper in this area could make a strong contribution. First, it could make a methodological contribution, which I think this paper claims. For example, it could propose a new method and show that it leads to better decisions or more reliable conclusions than existing approaches. More concretely, this could mean yielding better prediction of human scores or identifying biases more efficiently than existing methods do, both of which require a reasonably substantial amount of empirical evidence. In the current version, however, the paper primarily reads to me as a demonstration of how to apply standard Bayesian GLM tools to this setting, with one additional real-data illustration. Therefore, I do not see enough evidence to view this as a significant methodological contribution.
> >
> > Second, it could make an empirical contribution by uncovering new and well-supported insights about LLM-as-a-judge systems. Here too I remain unconvinced. The paper is centered mostly around simulated examples, and the MT-Bench case study, while helpful, is relatively limited: it is confined to one dataset, appears only in the appendix, and does not provide the depth of empirical analysis I would expect for this venue. In particular, if the focus is new empirical insights on biases, I would hope to see validations on several real datasets and tasks, more comprehensive robustness analysis, and stronger discussion of the conclusions.
> >
> > For these reasons, I see the paper as closer to a useful framework/tutorial paper than a research work that reaches the threshold I would expect for acceptance here.
> >
> > 2. **I remain concerned about some of the paper's technical framing and claims.**
> >
> > I agree that using a predictive statistical model to quantify bias can be useful. My concern is with how strongly the paper interprets these quantities. In particular, the paper repeatedly emphasizes that the GLM framework enables "counterfactual simulations" by removing selected terms from the linear predictor and recomputing evaluation outcomes. As written, this goes beyond descriptive modeling and encourages a causal or intervention-style interpretation. I acknowledge that the paper does not make a full causal claim, but the rhetoric suggests that these simulations reveal what would happen if certain biases were removed. I do not think the paper establishes the assumptions needed for that interpretation, such as specification, omitted variables, etc.
> >
> > Relatedly, I find some of the presentation overconfident. For example, Appendix A.5 states that the framework extracts "all effects simultaneously from a single dataset," whereas the original MT-Bench paper relied on targeted controlled experiments for specific biases. I do not think these are interchangeable. A unified regression framework is elegant and potentially useful, but fitting an observational model and reading coefficients is not a substitute for identification through experimental design.
> >
> > 3. **The rebuttal on priors is helpful, but does not fully resolve my concern.**
> >
> > I appreciate the authors' clarification that they use weakly informative priors and that, in the MT-Bench experiment, tightening the priors (M3 vs. M6) did not significantly affect predictive fit or conclusions. This is a useful robustness check. However, I do not think it fully addresses the issue.
> >
> > My concern is not that the paper should prove priors are irrelevant, but that prior choice and sensitivity deserve more careful treatment given the paper's Bayesian framing and the interpretive claims built on top of the fitted model. Whether a prior is "weakly informative" is scale-dependent and depends on the design of predictors and link scale; it is not an intrinsic label. I also did not find the M3/M6 comparison documented clearly enough to evaluate exactly what changed and why. More broadly, if the paper wants to emphasize counterfactual-style simulations, then the sensitivity of those simulations to prior becomes even more important.
> >
> > Overall, I appreciate the authors' effort in the rebuttal and I agree the paper has become stronger. However, I remain unconvinced that the current version makes a sufficiently strong contribution for this venue. My score therefore remains unchanged.

---

> > > ### Author Response · Authors · 2026-04-04
> > >
> > > We thank the reviewer for acknowledging that we have addressed their main concerns about the data/empirical contribution (which were based on misreadings) and that "the paper is stronger". Given that these were the reviewer's strongest objections, we are surprised that the score remains unchanged. We clarify the reviewer's new concerns below, which should help with the remaining reservations:
> > >
> > >
> > > **About our contribution:** The reviewer mentions two possible contribution types (new method or new empirical findings) and argues we fit neither. We respectfully disagree. Many impactful papers contribute by bringing rigour to a domain where ad-hoc practices are widespread. The LLM-as-a-judge literature currently relies on isolated tests for individual biases, agreement metrics computed without uncertainty, and no principled way to distinguish systematic bias from fundamental disagreement.
> > >
> > > Our framework:
> > > - jointly models multiple bias sources with uncertainty quantification
> > > - enables model-based decomposition of inter-rater agreement
> > > - provides a unified workflow across ordinal, binary, and pairwise formats.
> > >
> > > To our knowledge, no prior work in this area has done this. The fact that the underlying tools are well-established in other fields is actually a strength: they have known properties, extensive documentation, and their implementation has been extensively validated.
> > >
> > >
> > > **About the framing:**
> > > - We see how the term "counterfactual simulation" could suggest causal identification, and we are happy to use another term. What we perform can also be seen as a sensitivity analysis in which we ask: "how would a summary statistic (e.g., Krippendorff's alpha) change if we removed a specific model component?". We are happy to clarify this in the paper.
> > > - We completely agree that controlled experiments provide stronger identification. Our point is not that GLMs replace experiments, but that they extract more from existing data, which is important in the many practical settings where re-running evaluations is infeasible. In fact, when controlled experiments are available, GLMs benefit even more, as cleaner data leads to more precise and reliable estimates.
> > >
> > >
> > > **About the priors:** We appreciate this feedback and will document the M3/M6 comparison more clearly. We also want to note that our framework does not have to be Bayesian: all models presented can equally be fit in a frequentist setup (e.g., using maximum likelihood estimation) as the core contributions (bias modelling, decomposition of disagreement, model comparison) are independent of the fitting method. The choice between Bayesian and frequentist inference is a well-established methodological decision with known tradeoffs, and researchers can choose whichever approach they prefer.

---

### Official Review · Reviewer_9EFv · 2026-03-13

**Soundness:** 2
**Presentation:** 4
**Significance:** 3
**Originality:** 2
**Overall Recommendation:** 4
**Confidence:** 3

**Summary:**

The authors study bias in LLM-as-a-judge evaluation. They argue that raw agreement statistics can be misleading because disagreement may reflect systematic grader effects rather than due to output quality. They propose a Bayesian GLM framework that uses various logistic models and hierarchical extensions to quantify biases in autograders. They claim that such a framework can jointly model multiple autograder biases directly from existing evaluation data and with uncertainty quantification. They claim the framework quantifies various elements of interest for autograders such as grader severity, self-bias, length bias, item-level effects, etc.

**Compliance With Llm Reviewing Policy:**

Affirmed.

**Final Justification:**

No rebuttal was submitted to my comments and so I maintain my score.

**Key Questions For Authors:**

1. Can the authors provide more real-world empirical analysis of the Bayesian GLM framework beyond the one done in the appendix?
2. Can the authors provide more comparisons with other statistical frameworks beyond their own?
3. The ordered logistic models seem to use the same cutpoints. Is that correct? This modeling assumption implies that all graders have the same rating scale, but in practice graders don't necessarily have the same harshness when evaluating and can compress their evaluation scales in other cases. Did the authors consider models with grader-specific cutpoints?

**Limitations:**

yes

**Strengths And Weaknesses:**

# Strengths
1. The paper is easy to read and very clearly motivated. It is clear what model is being fit and why the questions being asked are important.
2. The framework is quite broad and addresses a lot of open problems in LLM-as-a-judge
3. I think the paper is very practical and genuinely has a place in real-world analyses

# Weakness
1. The weakest point by far is that the novelty is not so great in my opinion. The authors propose the work as a "novel framework", but the pieces are quite standard statistical techniques. While there is merit to putting the pieces together, I think we need to ask what part of this is novel. The application is good to guide future researchers how to use these statistical tools to judge their LLM-As-A-Judge analyses, but the analysis itself does not actually offer as much novelty from a technical perspective.
2. The second weakest point is that the tests were all done on synthetic data, with the exception of the MT-Bench in the appendix. That is very problematic in my opinion. Furthermore, there are no comparisons to other statistical families of techniques. It would be good to see a comparison with other techniques outside of the framework.
3. I think some of the claims are not well supported in section 3. Self-bias and intransitivity were the weakest points in my opinion.

---

### Official Review · Reviewer_VbGS · 2026-03-13

**Soundness:** 3
**Presentation:** 2
**Significance:** 3
**Originality:** 3
**Overall Recommendation:** 4
**Confidence:** 2

**Summary:**

The proposed approach is a Bayesian GLM-based statistical framework that jointly models LLM evaluation scores and multiple sources of grader bias, quantifies them with uncertainty, and enables bias-removal simulations. This allows researchers to understand and quantify systematic errors in LLM‑as‑a‑Judge scenarios.

**Compliance With Llm Reviewing Policy:**

Affirmed.

**Key Questions For Authors:**

Is it possible that separate checks can be better than the unified approach proposed by the authors?
Is it possible to explain when the method is applicable?

**Limitations:**

A discussion about how this technique should still be gamed by LLMs if adopted, and re-acknowledging issues of LLJ for the impact statement could be helpful

**Strengths And Weaknesses:**

The paper is actually nice and interesting to read, and tackled a clear and timely problem with the rise of adoption of LLM-as-a-judge evaluations, but I do have some issues following without examples, which could have been helpful (maybe to be added in the appendix?). Quantifying multiple judge biases is a useful and fairly novel framing for LLM-as-a-judge evaluation. Many core demonstrations are simulation-based, with the main real-data validation (MT-Bench case study + model comparison) largely in the appendix. It would have been interesting to use concrete examples from the dataset (even just in the appendix. I actually couldnt follow the experiment in the appendix), to show the various aspects and how they were evaluated. Additionally, estimating certain biases (e.g., length) requires sufficient variation and overlap, which isn’t guaranteed in all pipelines, thus a short, explicit “applicability checklist” (overlap requirements, minimum data per interaction cell, diagnostics for confounding) would make it easier for practitioners to know when to trust the outputs.
Also, the paper emphasizes avoiding re-evaluation via counterfactual simulation, but lacks a concrete cost discussion.

---

### Decision · Program_Chairs · 2026-04-30

**Decision:**

Accept (regular)

**Comment:**

The paper proposes a method based on bayesian generalized linear models to quantify and mitigate LLM judge biases. The capabilities of the framework to provide uncertainty estimates, clarify sources of disagreement between judges, and perform counterfactual simulations without costly re-evaluation are demonstrated through simulated examples in the paper. The paper performs experiments on simulated data and a real dataset, albeit relegated to the appendix.

Multiple reviewers agreed that the paper is well-written and clearly motivates the problem. The paper’s claim of novelty was questioned by some reviewers, as the pieces put together in the framework are standard statistical techniques, however, other reviewers pointed out that these pieces are integrated in a unified, flexible, and principled framework that is valuable for LLM judge evaluation. For example, in a rebuttal, the authors pointed out that the models can also be fit in a frequentist setup, which I agree with, and should be expounded upon more in the paper to demonstrate the flexibility of the framework. The experiments were done on synthetic data with the exception of MT-Bench in the appendix which should be moved into the paper itself.